**COMMUNICATIONS**

# Quantitative characterization of 3D bioprinted structural elements under cell generated forces

Cameron D. Morley[1,9], S. Tori Ellison[2,9], Tapomoy Bhattacharjee[3], Christopher S. O'Bryan [1], Yifan Zhang [1], Kourtney F. Smith[2], Christopher P. Kabb [4], Mathew Sebastian[5], Ginger L. Moore [6], Kyle D. Schulze [7], Sean Niemi[1], W. Gregory Sawyer[1,2], David D. Tran[5], Duane A. Mitchell[6], Brent S. Sumerlin[4], Catherine T. Flores[6] & Thomas E. Angelini[1,2,8]

With improving biofabrication technology, 3D bioprinted constructs increasingly resemble real tissues. However, the fundamental principles describing how cell-generated forces within these constructs drive deformations, mechanical instabilities, and structural failures have not been established, even for basic biofabricated building blocks. Here we investigate mechanical behaviours of 3D printed microbeams made from living cells and extracellular matrix, bioprinting these simple structural elements into a 3D culture medium made from packed microgels, creating a mechanically controlled environment that allows the beams to evolve under cell-generated forces. By varying the properties of the beams and the surrounding microgel medium, we explore the mechanical behaviours exhibited by these structures. We observe buckling, axial contraction, failure, and total static stability, and we develop mechanical models of cell-ECM microbeam mechanics. We envision these models and their generalizations to other fundamental 3D shapes to facilitate the predictable design of biofabricated structures using simple building blocks in the future.

[1] University of Florida, Herbert Wertheim College of Engineering, Department of Mechanical and Aerospace Engineering, Gainesville, FL 32611, USA. [2] University of Florida, Herbert Wertheim College of Engineering, Department of Materials Science and Engineering, Gainesville, FL 32611, USA. [3] Princeton University, Department of Chemical and Biological Engineering, Princeton, NJ 08540, USA. [4] University of Florida, George and Josephine Butler Polymer Research Laboratory, Center for Macromolecular Science and Engineering, Department of Chemistry, Gainesville, FL 32611, USA. [5] Division of Neuro-Oncology, Preston A. Wells, Jr. Center for Brain Tumor Therapy, Lillian S. Wells Department of Neurosurgery, University of Florida, Gainesville, FL 32611, USA. [6] University of Florida, Brain Tumor Immunotherapy Program, Preston A. Wells Jr. Center for Brain Tumor Therapy, Lillian S. Wells Department of Neurosurgery, Gainesville, FL 32611, USA. [7] Auburn University, Department of Mechanical Engineering, Auburn, AL 36849, USA. [8] University of Florida, Herbert Wertheim College of Engineering, J. Crayton Pruitt Family Department of Biomedical Engineering, Gainesville, FL 32611, USA. [9] These authors contributed equally: Cameron D. Morley, S. Tori Ellison. Correspondence and requests for materials should be addressed to T.E.A. (email: t.e.angelini@ufl.edu)

While the creation and maintenance of multicellular structures with stable shapes is essential to tissues, organs, and engineered cell-assemblies, their mechanical deformations are often critical to proper development and function; these deformations can even arise in the form of mechanical instabilities like buckling[1–3]. Proliferating cells in the developing gut, for example, generate outward pressure that is more easily accommodated by undulations than compression or stretch[4–6]. Cell contraction can also generate mechanical instabilities in vitro; tensed fibroblasts wrinkle thin elastomer sheets, while cooperatively contracting cardiomyocytes can bend and buckle macroscopic objects[7–9]. These demonstrations of contraction-driven instability were enabled by the ability to design and fabricate substrates for careful in vitro study and indicate that cell-generated tension may also drive instabilities in 3D milieus, including biofabricated structures. Moreover, all these biomechanical behaviors arise in high aspect-ratio systems that approximate classic structural elements like beams, plates, and tubes. Thus, employing simple geometric elements and structural engineering principles in 3D biofabrication strategies may enable predictive and controlled design of dynamic multicellular assemblies in regenerative medicine and tissue engineering applications. Biofabrication technology for making living structural elements from only cells and extracellular matrix (ECM) is becoming increasingly available[10–14], however, the fundamental principles controlling the mechanical behaviors of such structural elements under cell-generated forces have not been established. Numerous models have been developed to reproduce individual observations of tissue deformation and instability[4,15], yet basic physical principles and simple mathematical relationships are critically needed to enable researchers to predict the behaviors of engineered 3D tissue elements.

Here, we investigate the mechanics of living structural elements, leveraging a 3D bioprinting method that enables their design, fabrication, and testing[14,16]. Microbeams made from cells and ECM are 3D printed within a growth medium made from packed microgels, which gently cradles the microbeams, provides a mechanically tuneable environment, and enables methodical studies of collective cell mechanics in 3D. To systematically test the variables controlling cell–ECM microbeam mechanics, we vary cell density, ECM concentration, microbeam diameter, and the surrounding medium material properties (Fig. 1). We find a cascade of cell-driven behaviors including beam buckling, break-up, and axial contraction. By modifying classic mechanical theories, we uncover basic principles of tissue microbeam mechanics that can be generalized to diverse cell types, ECMs, and bioprinting support materials. These foundational principles can be extended to other shapes such as sheets and tubes, enabling a component-oriented future of mechanical design in tissue engineering and biofabrication in which stability and instability are programmed into the tissue maturation process.

## Results

**Microbeam fabrication.** To investigate how cell-generated forces collectively drive shape changes in multicellular structures, we 3D print cell–ECM mixtures into a jammed microgel medium, leveraging its yielding properties. The 3D printing and culture medium is created by swelling microgels in liquid cell growth media. Fibroblast (3t3), glioblastoma (GL261), and pancreatic cancer (Panc02) cells are cultured in 2D, harvested, mixed with collagen-1 solution, and loaded into syringes. While collagen-1 does not recapitulate the ECM these cells encounter naturally, they can attach to the matrix and contract (Supplementary Movie 1). The syringe needle is inserted into the microgel medium and translated while injecting cell–ECM mixtures, creating microbeams of diameter 50–200 μm (Fig. 2a, Supplementary Fig. 1). This approach provides a pliable environment that enables quantitatively testing the mechanical behaviors of cell–ECM structures (see Methods for microgel synthesis and sample preparation details).

**Microbeam buckling wavelength measurement and analysis.** Confocal microscopy images reveal that within 30 min after printing, the collagen-1 polymerizes while printed microbeams remain straight (Figs. 1b, 2a, Supplementary Fig. 2). After 24 h, the beams exhibit undulations having wavelength, $\lambda$, that varies with beam radius, $R$ (Fig. 2, Supplementary Movie 2). We measure $\lambda$ using multiple different approaches and analyze the $\lambda$—$R$ trend using a relationship from Euler–Bernoulli (EB) theory of beam buckling inside an elastic continuum (Supplementary Figs. 3 and 4). In EB beam theory, the shape of a beam embedded in an elastic medium is described by the equilibrium balance of forces, given by

$$EI\frac{d^4x}{dz^4} + F\frac{d^2x}{dz^2} + G'x = 0, \qquad (1)$$

where $x(z)$ is the lateral deflection of the beam at location $z$ along its backbone, $E$ is the elastic modulus of the beam, $F$ is the force

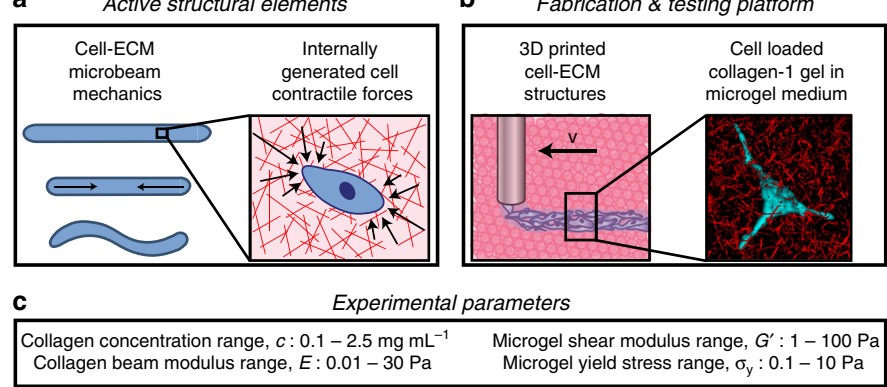

**Fig. 1** Packed microgels provide a mechanically tuneable environment for methodical studies of collective cell mechanics in 3D. **a** Analogous to externally applied loads in classical beam mechanics, internal forces generated by contracting cells drive the undulation of microbeams made from ECM. **b** Fabricating cell–ECM microbeams is performed by 3D printing into a cell culture medium made from jammed microgels. This soft environment provides mechanical support to extremely delicate beams while simultaneously facilitating macroscale deformations driven by cell contraction. **c** We systematically investigate cell-driven mechanical behaviors by varying the properties of ECM microbeams and the surrounding microgel medium over the ranges given here

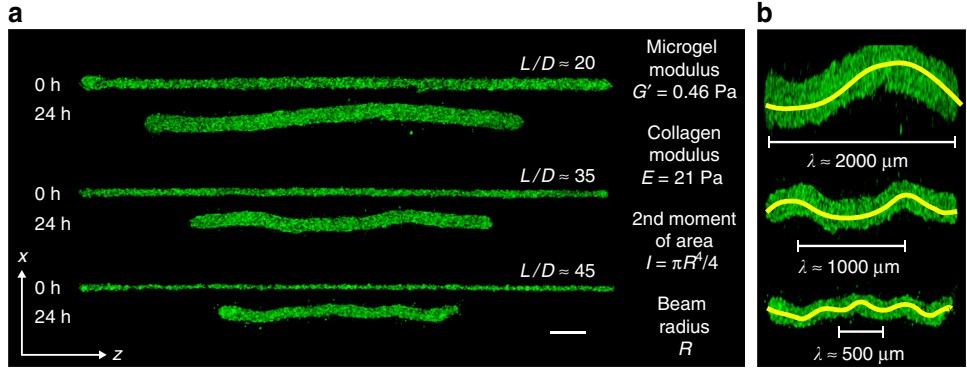

**Fig. 2** 3D printed microbeams made from cells and ECM appear to buckle. **a** 3D printed cell–ECM microbeams having lengths $L = 5$ mm and varying diameters, $D$, develop undulations over a 24 h period (Scale bar: 250 μm). **b** Digitally stretched images of beams from (**a**) accentuate the undulations and reveal a relationship between beam radius, $R$, and wavelength, $\lambda$ (lines manually drawn)

applied along the $z$-axis, and $G'$ is the shear modulus of the surrounding medium. $I$ is the second moment of area, given by

$$I = \frac{\pi R^4}{4},\qquad(2)$$

for a beam of circular cross-section and radius, $R$. Since this is an equilibrium equation, it applies to situations where inertia is negligible and $F$ is a constant, balanced locally at each infinitesimal element by the internal costs to bend the element and the external cost to deform the medium surrounding the element. The solution to this equilibrium equation is sinusoidal, and the lowest energy shape has a wavelength given by

$$\lambda = 2\pi \left(\frac{EI}{G'}\right)^{1/4}.\qquad(3)$$

To test whether this relationship applies to cell loaded ECM microbeams, we independently measure $\lambda$, $E$, $I$, and $G'$ for numerous beams of different compositions embedded in multiple different formulations of microgel media.

To predict our measurements of $\lambda$, we determine $G'$ with a rheometer; $E$ is more challenging as the elastic moduli of collagen networks in shear, compression, and tension strongly differ and remain under investigation[17–19]. To determine $E$ within a beam-buckling context, we 3D print cell-free collagen-1 beams into the microgel medium and measure their responses to manually applied axial loads. We observe buckling with macro-scale beams of diameter 0.5–2 mm and micro-scale beams of diameter 50–200 μm. Measuring $\lambda$ and $R$, we determine $E$ for beams printed at different collagen concentrations. To test whether EB theory applies to cell–ECM microbeams, we plot measurements of $\lambda$ versus $2\pi\,(EI/G')^{1/4}$, varying $R$ and $E$, using all three cell types. We find that EB theory predicts our data with no fitting parameters ($R^2 = 0.93$); control experiments without cells exhibit no spontaneous buckling. Thus, cell–ECM microbeam undulations are a form of buckling driven by the cells within (Fig. 3, Supplementary Figs. 5–8). The dependence of $\lambda$ on $G'$ indicates that the microgel pack is deformed elastically and not yielded during buckling. However, if microgel creep occurs over the long time-scale associated with the process, we expect the slow rearrangement of the microgels to occur at constant packing density[20].

It is noteworthy that the one-fourth power makes the wavelength less sensitive to $E$ and $G'$ than to $R$. For example, a 100% error in $E$ or $G'$ results in 19% error in $\lambda$; by contrast, errors in $\lambda$ are linearly proportional to $R$. Our confidence in $G'$ and $R$ measurements are very high, while our method of determining $E$ was developed for this manuscript and carries more uncertainty.

One source of uncertainty in measuring $E$ is the change in beam volume that occurs from the time of fabrication to the time of measurement. We account for this change by rescaling the estimated collagen concentration at the time of measurement based on the volume change from the time of fabrication. However, the added uncertainty from this procedure and the variability in repeated measurements of $E$ are relatively low compared to the mean values and to the overall range measured (Fig. 3c).

**Critical stress for microbeam buckling.** Since EB beam theory predicts $\lambda$, we extend this analysis to predict the load cells must generate to buckle the beams they reside in. To find the critical force required to buckle a beam, the lowest energy solution is substituted into the equilibrium equation and $F$ is solved for, yielding

$$F_{\mathrm{b}} = \frac{4\pi^2}{\lambda^2}EI + \frac{\lambda^2}{4\pi^2}G',\qquad(4)$$

where $F_{\mathrm{b}}$ is the critical buckling force. Substituting the buckling wavelength formula into the critical force equation, and recognizing that buckling occurs when the cost to bend the beam and the cost deform the surrounding medium are comparable, a simplified formula for the critical force is found, given by

$$F_{\mathrm{b}} \approx R^2\sqrt{\pi E G'}.\qquad(5)$$

Dividing $F_{\mathrm{b}}$ by the beam cross sectional area, we write down the critical stress applied to each element of the beam, given by

$$\sigma_{\mathrm{b}} \approx \sqrt{\frac{EG'}{\pi}}.\qquad(6)$$

We observe no bending in cell-free microbeams, so we approximate the externally applied force to be zero. Since undulations are observed with cell-loaded microbeams, we treat $F$ as an average cell generated force, acting to compress and bend the collagen microbeam while also deforming the surrounding microgel medium. Accordingly, $\sigma_{\mathrm{b}}$ is the averaged-out cell-generated stress acting to deform the beam and the surrounding medium. We use this relationship to determine single-cell generated stresses in collagen-1 microbeams, later in the manuscript.

**Increasing $G'$ to suppress buckling leads to beam failure.** Our prediction of $\sigma_{\mathrm{b}}$ from EB beam theory indicates that increasing $G'$ of the microgel medium will eliminate buckling. To explore the potential for $G'$ to control beam response to internally generated cell contraction, we 3D print cell–ECM microbeams within

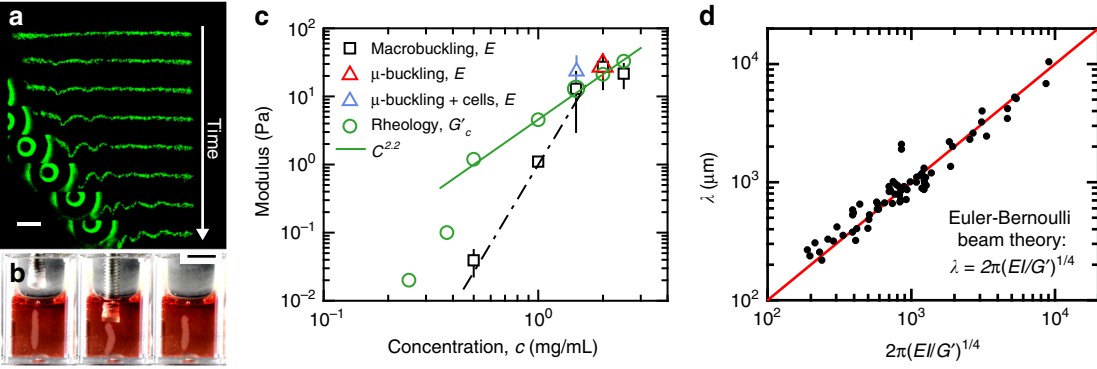

**Fig. 3** Determining the microbeam moduli to predict $\lambda$. **a** Microbuckling: we mount our 3D printer atop an inverted epifluorescence microscope and print horizontal beams made from 2.0 mg/mL collagen-1 supplemented with 1 μm fluorospheres to enable imaging. After gelation, we apply a large axial load to the beam, observing clear transverse undulations. **b** Macrobuckling: we 3D print vertical beams made from 0.5–2.5 mg/mL collagen-1 and load them axially, also observing buckling. (Scale bar: 5 mm). **c** For both macrobuckling and microbuckling tests, we determine the beam modulus, $E$, from EB theory. For comparison, we plot $G'$ of collagen networks measured with shear rheology. The solid green-line agrees with previous reports on collagen rheology; the dashed line is used for estimating $E$ between measured datapoints (0.5 and 1.5 mg/mL collagen). Errorbars are ±one standard deviation. **d** Euler–Bernoulli (EB) theory predicts the relationship between the beam elastic modulus, $E$, the supporting material shear modulus, $G'$, the second moment of area, $I$, and the wavelength, $\lambda$. With no fitting parameters, measurements of $\lambda$ from many different beams are predicted ($R^2 = 0.93$). ($n = 69$ separate measurements displayed. See Supplementary Fig. 7 for breakdown by cell type.)

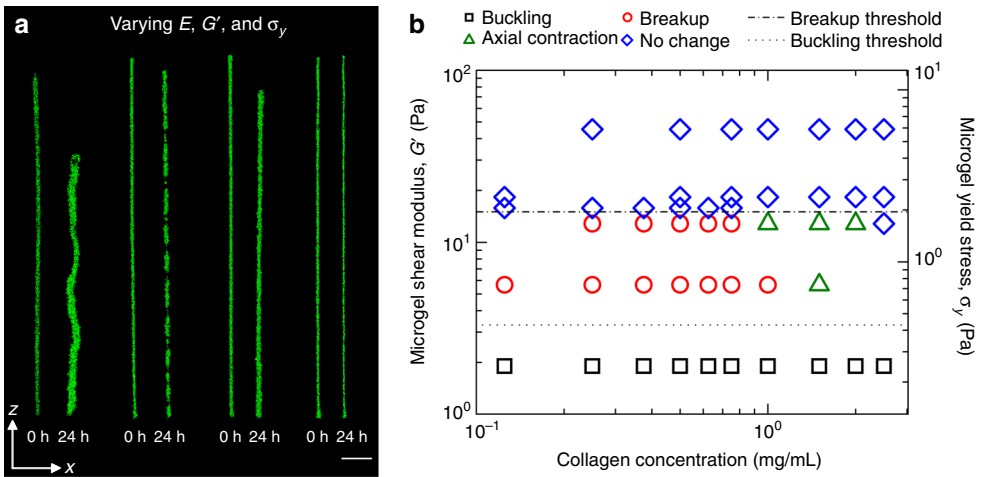

**Fig. 4** Microbeam mechanical behaviors controlled by beam and microenvironment material properties. **a** By varying the shear modulus of the surrounding microgel medium, $G'$, and the beam elastic modulus, $E$, we observe a cascade of different behaviors; the cell–ECM microbeams buckle, breakup, contract axially, and remain stationary. (left to right: collagen $E = 0.035$ Pa and microgel $G' = 1.92$ Pa; $E = 0.1$ Pa, and $G' = 5.69$ Pa; $E = 1$ Pa and $G' = 10.85$ Pa; $E = 0.3$ Pa and $G' = 55.02$ Pa. Scale bar: 1 mm). **b** A two-dimensional map of these behaviors illustrates where transitions occur. Dotted line indicates $G'^b$ and dashed line indicates $\sigma_y^f$. ($n = 3$ samples observed for each displayed data point.)

microgel media prepared with higher $G'$. We find that buckling is eliminated, and instead the microbeams break into small contracting segments. Thus, at a level of internally generated stress, $\sigma_{int}$, a threshold shear modulus of the microgel medium is predicted to be $G'^b \approx \pi\sigma_{int}^2/E$. Constructing a stability diagram, we find a threshold value for $G'^b$ of 3.4 Pa. The independence of $G'^b$ with collagen concentration suggests the cell-generated stress within microbeams is proportional to $E^{1/2}$, which we explore later (Fig. 4, Supplementary Movie 3).

To aid investigation of the observed microbeam break-up at increased $G'$, we develop a model following classical failure analysis. The break-up of cell–ECM microbeams does not appear to satisfy the assumptions of classical failure models like Griffith's theory of brittle materials failure or ductile failure. For example, no clear cracks are detectable, and the yielding threshold appears to be dominated by the yield-stress of the surrounding microgel medium rather than the material properties of collagen network (Fig. 4b). We considered the possibility that beam break-up is

limited by a form of friction between collagen fibers and microgels at the beam surface. However, in such a case we would expect to observe a dependence of the failure threshold on the collagen concentration, which would control the strength of the interface. The failure threshold appears independent of collagen concentration, so to further investigate the lack such a trend, we collected 3D images of the interface between a collagen beam and the surrounding microgel medium using confocal fluorescence microscopy. In these images we observe a strikingly circular beam cross-section and a zone of intermixing between the microgels and the collagen approximately 25 μm in thickness (Supplementary Fig. 9). Taken together, our observations indicate that the surfaces of contracting beams drag intermixed microgels axially, limited by the stresses associated with microgel–microgel sliding and flow just outside the intermixed zone. It is also interesting to consider whether cell-driven remodeling of collagen fibers at the beam surface plays a key role in the break-up process. While investigating these detailed microscopic dynamics would

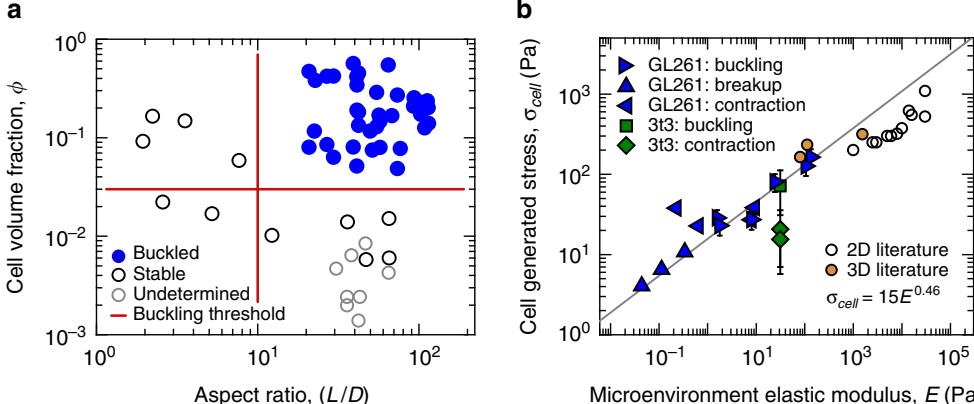

**Fig. 5** Determining cell generated stress from instability thresholds and axial contraction. **a** We tune the average cell-generated stress within microbeams by varying cell volume fraction, $\phi$, at constant collagen concentration (2 mg/mL) and microgel shear modulus (0.46 Pa). Beams with $\phi < 0.03$ do not buckle; data from sparse beams are marked undetermined as buckling is not observed but difficult to rule-out. Low aspect ratio beams ($L/D < 10$) remain straight. (separate measurements displayed: $n = 36$ buckled, $n = 11$ stable, and $n = 8$ undetermined data points.) **b** We estimate the average stress generated by single cells, $\sigma_{cell}$, in beams that buckle, break-up, and contract. Most data points follow a scaling law relating $\sigma_{cell}$ to $E$ that nearly extrapolates to 2D and 3D traction force microscopy data. Error bars correspond to ±standard deviation of repeated measurements. (separate measurements displayed: $n = 5$ GL261 buckling, $n = 3$ GL261 breakup, $n = 4$ GL261 contraction, $n = 1$ 3t3 buckling, $n = 2$ 3t3 contraction; literature data: $n = 11$ 2D, and $n = 3$ 3D.)

elucidate how interfacial interactions may contribute to large-scale beam behavior, such studies would entail thorough experimentation outside the range of approaches taken here.

We, therefore, develop a failure model that balances the internal stress built up within the collagen microbeam before failure against the yield stress of the microgel material that appears to set the break-up threshold. Following classical methodologies, we compute the total strain energy within the beam before failure, given by

$$U_s \approx \frac{\sigma_{int}^2}{E} \pi R^2 L_0, \tag{7}$$

where $\sigma_{int}$ is the internal stress level, $E$ is the beam elastic modulus, $R$ is the beam radius, and $L_0$ is the beam length. When the beam breaks, the flow of microgel material into the space between separating segments comes at an energetic cost, resisting the motion of separating segments. Thus, we estimate the total energy to break the beam into $N$ segments of length $L_1$ to be

$$U_y \approx \sigma_y N \pi R^2 L_1, \tag{8}$$

where $\sigma_y$ is the yield stress of the surrounding medium; here we assume that a hydrodynamic volume of microgel material is yielded around the segments equal to the segment volume. Equating these energies to find the threshold stress, and empirically recognizing that $N L_1 \approx L_0$, we find

$$\sigma_{int} \approx \sqrt{E \sigma_y}. \tag{9}$$

Interestingly, this form is similar to the result found using Griffith's criterion if the effective surface energy density is given by $\gamma \approx R\sigma_y/2$ and the crack-length is the radius of the beam. Substituting this effective surface energy density into Griffith's criterion creates a slightly different prediction for the conditions under which our microbeams will fail, given by

$$\sigma_{int} = \sqrt{\frac{E \sigma_y}{\pi}}. \tag{10}$$

While it may be instructive to consider how the yield stress of the surrounding medium creates an effective surface energy density, we leave this comparison for future work that will elucidate the details of how extremely weak structures fail while embedded in stronger surroundings. As with the EB beam theory analysis, above, we treat the origin of the internal beam stress as

cell contraction to determine the level of cell-generated stress by identifying the threshold values of $\sigma_y$ and $E$ for microbeam break-up (Fig. 4b). At this threshold, we predict $\sigma_f \approx \sqrt{\sigma_y E}$, where $\sigma_f$ is the applied stress at failure, indicating that increasing $\sigma_y$ of the microgel medium will eliminate break-up. Correspondingly, for a given a level of internally generated stress at failure, $\sigma_f$, the threshold $\sigma_y$ for beam failure is given by $\sigma_y^f \approx \sigma_f^2/E$. Like the buckling threshold, this failure threshold appears to be independent of collagen concentration and occurs at $\sigma_y^f = 1.95$ Pa, indicating that $\sigma_f \sim E^{1/2}$. Microbeams in microgel media with $\sigma_y > 1.95$ Pa remain stable, straight, and intact throughout the 24-h tests. Thus, when the microgel yield stress is high enough, the cells cannot generate enough stress to flow the microgels into potential open spaces, remaining intact at all collagen concentrations.

We hypothesize that cells sense the collagen network elastic modulus in their microenvironments to set the stress they apply to microbeams. Accordingly, by equating $\sigma_b$ and $\sigma_f$ in the buckling and failure models at the same values of $E$ predicts $G'^b = \pi \sigma_y^f$; our experiments show that $G'^b \approx 1.65 \sigma_y^f$, less than a factor of two from the prediction (dashed and dotted lines in Fig. 4b). The failure model breaks down at high-ECM concentrations, where beams contract axially without failing. This behavior suggests that at low-collagen concentrations, the microgel medium primarily resists beam breakup; at high concentrations the ECM network controls beam integrity. In this regime, beams contract axially by 1–5% (Figs. 4b, 5, Supplementary Movie 4). We perform additional experiments on these contracting beams to further test the potential for network remodeling and the presence of cells to alter the collagen gel properties. We manually apply axial loads to drive these beams to buckle, enabling their elastic moduli to be determined by measuring their buckling wavelengths. We find that at both $t = 0$ h and 24 h, manual loading causes the beams to buckle. The measured wavelengths and estimated moduli determined from these tests agree with our predictions of $\lambda$ and measurements of $E$ performed in the cell-driven and cell-free buckling tests, described earlier (Fig. 3, Supplementary Figs. 6 and 7). We also measure buckled, contracted, and stable beams at the 48 h time-point, finding no relaxation or transition between different classes of deformed state relative to the 24 h time-point, indicating any

degradation or remodeling of the collagen network is insufficient to disrupt the deformed state of the beams. As an additional test on these contracting beams, we considered that a surface-area dependent friction force may limit contraction; more force is required to pull a long rope through a gripping tube than a short rope. Indeed, we find that shorter beams 1 mm in length contract yet longer beams 30 mm in length do not contract. Thus, a friction-dominated limit appears to emerge with increasing beam length (Supplementary Fig. 10).

**Stabilizing microbeams and estimating single-cell stresses**. In the models employed here, we treat the internally generated stress as the averaged strain energy per unit volume generated by single cells, given by $\phi\sigma_{cell}$, where $\phi$ is the cell volume fraction and $\sigma_{cell}$ is the amount of stress a single cell generates within the ECM microbeam. Thus, by varying cell density, the average internal beam stress can be tuned to control instabilities. In addition, low aspect-ratio structures should suppress buckling. To further test the applicability of EB theory, we 3D print numerous beams containing different cell volume fractions, $\phi$, and different aspect ratios, $L/D$, where $L$ and $D$ are beam length and diameter. To isolate the effects of cell density, all these beams are prepared at a collagen concentration of 2 mg/mL and printed into microgel medium with $G' = 0.46$ Pa and $\sigma_y = 0.06$ Pa. Under these conditions, we find beams loaded with cells below $\phi = 0.03$ and $L/D = 10$ do not buckle (Fig. 5a). We note that most tissues constitute cells at high packing fractions exceeding the hard-sphere random close-packing fraction, $\phi \approx 0.64$. We tested beams up to packing fractions approaching these levels ($\phi \approx 0.6$), which exhibit buckling. However, we limit the detailed analysis shown in Figs. 2c, 4b, and 6b to beams having packing fractions less than $\phi \approx 0.2$ to avoid the potentially large errors associated with dramatically different collagen network structures that must occur at high volume fractions. To quantitatively study 3D printed structures at high volume fractions, new approaches to measuring beam elastic moduli need to be developed (Supplementary Fig. 11).

To estimate the level of stress that single cells generate within the microbeams having $\phi \leq 0.2$, we use the measured threshold stress values required for beam buckling and failure. Thus, the single cell stress at the buckling threshold is given by

$$\sigma_{cell} \approx \frac{1}{\phi}\sqrt{\frac{EG'^{b}}{\pi}}, \qquad (11)$$

and the single-cell stress at the failure threshold is given by

$$\sigma_{cell} \approx \frac{1}{\phi}\sqrt{E\sigma_y^{f}}. \qquad (12)$$

By identifying the values of $\phi$, $E$, $G'$, and $\sigma_y$ at which buckling and failure thresholds are observed, we determine the level of single cell stress applied to the microbeams from within. Using data measured at these thresholds (Fig. 4b for GL261 and 6a for 3t3), we construct a plot of single-cell generated stress versus $E$, including both buckling and breakup data-points. A best-fit scaling law, $\sigma_{cell} = 15E^{0.46}$, overlays the data-points very well (Fig. 5b, $R^2 = 0.96$).

Cell generated stress can also be estimated by analyzing stable beams that contract axially; within a small window of conditions, cell-loaded ECM microbeams contract axially by 1–5%. This window is bounded by the microgel medium properties: on the low-end by the buckling threshold elastic modulus, $G' = 3.4$ Pa; on the high-end by the break-up threshold yield stress, $\sigma_y = 1.95$ Pa. Within these limits, the contraction window is also bounded by microbeam collagen concentration; below 1–1.5 mg/mL, the microbeams break-up; above this concentration, the microbeams contract while remaining straight. This threshold collagen concentration range corresponds to a collagen elastic modulus range of 1–10 Pa. Given the low level of axial contraction, we estimate the beam strain, $\varepsilon$, from its fractional change in length, given by

$$\varepsilon = \frac{\Delta L}{L}, \qquad (13)$$

where $L$ is the beam length right after printing and $\Delta L$ is the observed change in length. Thus, the corresponding stress in the beam is approximately the product of the beam elastic modulus and this strain, given by $\varepsilon E$. In addition, axial contraction is

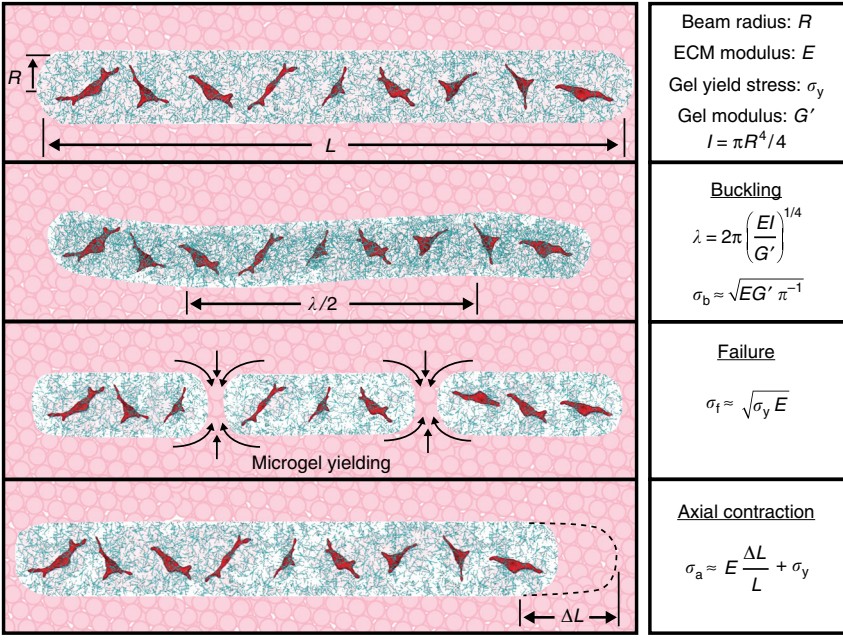

**Fig. 6** Summary of cell–ECM microbeam mechanics. By varying microbeam parameters and the material properties of the microgel medium, we observe a series of transitions between straight beams (top panel), buckled beams (second to top panel), broken beams (second to bottom panel) and axially contracted beams (bottom panel)

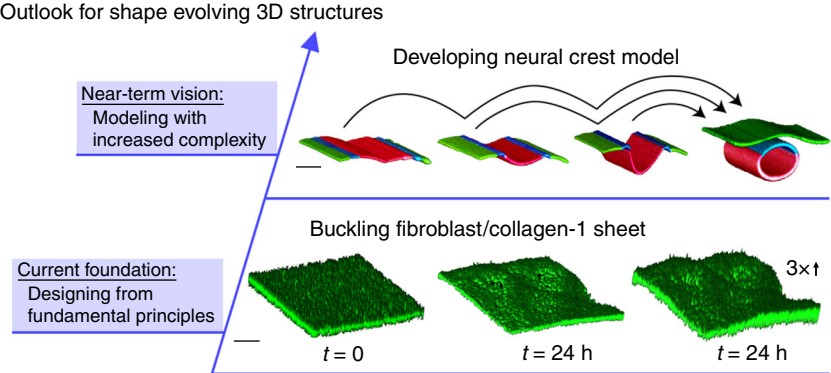

**Fig. 7** Outlook for cell-driven and shape-changing 3D structures. To expand our approach beyond the microbeam, we created sheets made from cells and collagen-1, finding that they too exhibited a buckling instability (bottom row). Confocal micrographs show a flat sheet immediately after printing (left) and a buckled sheet 24 h after printing (middle; right image stretched 3× vertically to accentuate undulations. Scale bar: 500 μm). We hope to one day create 3D versions of textbook-level models of developing tissues from multiple cell types. We demonstrate the readiness of the fabrication technique by printing static models of a neural crest tube at different stages of development, made of microspheres that fluoresce in three different colors. Confocal micrographs show the four structures, all made from three different materials (Scale bar: 2 mm; See Supplementary Information on 3D printed neural crest model)

resisted by the surrounding microgels, which will slowly yield and re-arrange as the beam contracts. Accounting for this additional stress required for cells to drive beam axial contraction, we set up an equilibrium equation given by

$$\sigma_{cell}\phi = \varepsilon E + \sigma_y, \qquad (14)$$

where cellular contractile stresses are resisted by the collagen beam elasticity and the microgel yield stress. The low levels of strain observed here, combined with the low-elastic moduli of collagen used in these samples, correspond to stress levels between 0.01 and 0.5 Pa; by contrast, the yield stress of packed microgel medium used here is between 0.44 and 1.95 Pa. In every case measured, we find that the elastic contribution to equilibrium is negligible compared to the yield stress of the surrounding microgel pack. Employing this model of axial contraction in combination with our failure model, we test whether the threshold between these two behaviors can be predicted. In the stability diagram displayed in Fig. 4b, the point at which the threshold yield-stress for breakup meets the threshold collagen concentration for axial contraction occurs at $\sigma_y = 1.95$ Pa and $E = 1$ Pa (1 mg/mL collagen concentration). Equating the internal stress from the two models at this triple-point and recognizing that $\varepsilon E$ is negligible compared to other terms, we predict that $E = \sigma_y$ at this point, within a factor of two of the observed location of the point. We summarize all the mechanical models explored here in Fig. 6.

To examine how our 3D estimates of $\sigma_{cell}$ compare to their 2D counterparts, we surveyed the literature reporting traction-force microscopy measurements. While cells in 2D are cultured on much stiffer substrates and cell-generated stresses are correspondingly larger, our results extrapolate to the established 2D measurements to within about a factor of two[21–23]. In addition, recent investigations of cell-generated traction forces in 3D matrices agree very well with our measurements, laying close to the extrapolated fit to our data-points[24–26]. In the 3D cases, we determined cell generated stress from reported strain-energy and estimated cell volume. Taken together, these results suggest a possible universal scaling relationship between single cell-generated stress and micro-environmental elastic modulus, consistent with our observations that $\sigma_{cell}$ scales like $E^{1/2}$ (Fig. 5b and Supplementary Fig. 12).

## Discussion

Mechanical instabilities occur throughout the body at multiple length-scales and stages of life. At large scales, arteries buckle, skin wrinkles, and the growing brain develops deep folds[27–30]. At smaller scales during development, multicellular epithelial folding and other collective motions coordinate with signaling events[31,32], while cell contraction, shape change, and proliferation create stress gradients that produce rugose surfaces and writhing tubes[3,6,33]. Controllably facilitating such structural changes in vitro remains a major challenge in engineering tissues and organs. Scaffolds provide predefined structures to guide cell assembly[34,35], but the conflicts inherent to simultaneously providing nascent structure and latent plasticity have necessitated scaffolds with increasing complexity in their synthesis, processing, and implementation. To overcome these challenges we developed a bioprinting method that allows the design and fabrication of structures made from only living cells and natural ECM that can evolve in shape under cell-generated forces; we 3D print cell–ECM structures into a 3D culture material made from jammed microgels swollen in liquid growth media[14,16]. This medium has a low yield stress (0.1–10 Pa) and its granular structure allows multicellular assemblies to change shape while remaining supported in a mechanically well-defined environment. While the deformations and dynamics of the cell–ECM structures studied here do not mimic the extreme morphological changes that occur in development, they represent a starting point for understanding and controlling instabilities and mechanical behaviors of simple biofabricated elements.

In the work presented here, we focus on simple cell–ECM microbeam mechanics, yet the biofabrication technique is sufficiently precise and versatile for investigating instabilities of different fundamental shapes or more complex structures made from multiple cell types. For example, planar sheets made from collagen-1 and 3t3 fibroblasts exhibit buckling after 24 h, much like the beams explored in detail, above (Fig. 7). These sheets were made at a collagen concentration of 0.5 mg mL$^{-1}$ with a corresponding elastic modulus of 0.04 Pa, and were printed into microgel medium having a shear modulus of 1.92 Pa and a yield stress of 0.25 Pa. We chose these parameter values because microbeams biofabricated under the same conditions exhibit a buckling instability; we measure a buckling wavelength of $\lambda \approx 1$ mm for the cell–ECM sheet. Making sheets, tubes, and other simple elements from multiple cell types is possible; we print 3D models of textbook-level snapshots of the developing neural crest

made from fluorospheres having different emission colors. This multimaterial print is achieved by programming our printer to fill from different wells and print different sections of the structure, sequentially (Fig. 7; Supplementary Information on 3D printed neural crest model). We envision that in the near future the same structures will be made from living cells, enabling the progression between different stages of development to be investigated in which signaling and related biochemical factors can be controlled and measured. The basic principles of mechanical deformation and instability, established here, will facilitate such investigations.

The recent decades of progress in mechanobiology has shown that cells in 3D ECM and their 2D counterparts on culture plates differ in shape, cytoskeletal architecture, focal adhesion distribution, and migration behavior[36–38]. Considering these differences, we were surprised to find a relationship between cell-generated stress and microenvironmental elastic modulus that connects the two limits. However, commonalities exist: cells in 2D and 3D exhibit similar mechanical behaviors, performing cycles of adhesion, contraction and detachment while demonstrating sensitivity to ECM concentration. Contracting elements dispersed within polymer networks have been investigated by using molecular motors and cytoskeletal filaments in vitro[39,40]; we expect these active matter physics approaches to elucidate cell dynamics in the 3D microenvironments investigated here[41]. This effort will be facilitated by including the complex responses that ECM networks exhibit locally under cell-generated forces, observed in 3D traction force experiments and theory[18,24–26,36,37]. In the immediate term, we hope the basic mechanical principles discovered here will guide biofabrication efforts for tissue engineering and regenerative medicine applications. While we focused on structures having a low cell density, our experimental approaches and mechanical models may be applied to densely packed cellular structures too; the elastic modulus of a beam made from densely packed cells could be measured with our manual buckling method. Thus, we believe the basic principles of stability and instability established here will accelerate the process of building complex structures from predictable, simple parts in analogy to how macroscopic structures are engineered, but now at the small-scale using living materials.

## Methods

**Microgel synthesis and 3D media formulation**. Lightly cross-linked poly-acrylamide microgels with 17 mol% methacrylic acid as an ionizable comonomer are prepared[20,42]. A solution of 8% (w/w) acrylamide, 2% (w/w) methacrylic acid, 1% (w/w) poly(ethylene glycol) diacrylate (MW = 700 g mol$^{-1}$), and 0.1% (w/w) azobisisobutyronitrile in ethanol (490 mL) is prepared. The solution is sparged with nitrogen for 30 min, then placed into a preheated oil bath set at 60 °C. After approximately 30 min, the solution becomes hazy and a white precipitate begins to form. The reaction mixture is heated for an additional 4 h. At this time, the precipitate is collected by vacuum filtration and rinsed with ethanol on the filter. The microparticles are triturated with 500 mL of ethanol overnight. The solids are again collected by vacuum filtration and dried on the filter for ~10 min. The particles are dried completely in a vacuum oven set at 50 °C to yield a loose white powder. The purified microgel powder is dispersed in cell growth media at various concentrations and mixed at 3500 rpm in a centrifugal speed mixer[14,16] in 5-min intervals until no aggregates are apparent. The microgel is then neutralized to a pH of 7.4 with NaOH and 25 mM HEPES buffer (Part no. BP299-100) and is left to swell overnight, yielding microgel 3D printing and growth media at concentrations of 2.2–10% (w/w).

**Cell culture and 3D printing preparations**. NIH-3t3 cells (murine fibroblast, ATCC CRL-1658) are cultured in Dulbecco's Modified Eagle Medium (DMEM) with 4.5 g/L glucose, L-glutamine, and sodium pyruvate supplemented with 10% FBS and 1% penicillin streptomycin. Glioblastoma cells (Glioma 261, NCI DCTD DTP C57BL/6) are culture in DMEM F12 with Glutamax, supplemented with 10% FBS and 1% penicillin streptomycin. Pancreatic cancer cells (PAN 02 NCI DCTD DTP 0507794) are cultured in RPMI 1640 with L-glutamine and sodium bicarbonate supplemented with 10% FBS and 1% penicillin streptomycin. When the cells have reached 70% confluence, they are dyed with cell tracker green (CMFDA) (Thermo-Fisher, part no. C2925), washed with PBS, and incubated in 3 mL of 5% Trypsin—EDTA solution for 5 min (GL261 cells express GFP and are not dyed. See

next section). The cells are harvested from the plate and placed into a 15 mL centrifuge tube, where they are centrifuged at 650g for 3 min. The supernatant is removed from the tube and bovine collagen-1 solution (Advanced BioMatrix, Part no. 5010-50 ML) is added. The cell pellet is dispersed with gentle pipette mixing and loaded into a 100-250 μL Hamilton gas-tight syringe. Finally, a sterile, blunt-tip 30 gauge luer-lock needle (SAI, part no. B30-50) is affixed to the syringe.

The microgel 3D printing and culture medium is prepared for each cell type using the corresponding liquid media. To enable the fabrication of microbeams having different diameters or lengths in separate wells within one sample preparation process, we typically use 12-well plates; single 35-mm petri dishes are used for single-beam prints (In Vitro Sciences, Part no. D35-10-0-N). To facilitate fluorescence imaging, glass bottomed vessels are always used, and multi-well plates with opaque walls are employed to eliminate light penetration into adjacent wells during imaging (Cellvis, part no. P12-1.5H-N). When 12-well plates are used, 1.5 mL of microgel media is loaded into each well. Prior to transferring to the printing stage, plates or dishes containing microgel media are incubated at 37 °C and 5% CO$_2$ for 1–2 h. During this incubation process, gas bubbles occasionally appear which are removed by centrifugation at 2000 rpm for 1 min.

**GL261-GFP reporter line generation**. GL261 (Glioma 261) cells were supplied by the National Cancer Institute (NCI) Division of Cancer Treatment & Diagnosis (DCTD) Developmental Therapeutics Program (DTP) Tumor Repository. To produce the glioblastoma cell line used here, lentivirus production is performed by polyethylenimine transfection of 293FT cells with pReceiver-Lv120-GFP (Genecopoeia EX-EGFP-Lv120) and helper plasmids pMD2.G and psPAX2 (Addgene plasmid 12259 and 12260). Viral supernatant is collected 48 h after transfection, filtered through a Stericup 0.45 mm filter and then concentrated by ultra-centrifugation. GL261 cells are plated in 6-well dishes at $5 \times 10^5$ cells per well and incubated with lentivirus in the presence of polybrene (8 mg/ml).

**Reporting summary**. Further information on research design is available in the Nature Research Reporting Summary linked to this article.

## Data availability

Data supporting the findings of this manuscript are available from the corresponding author upon reasonable request. A reporting summary for this Article is available as a Supplementary Information file.

The source data underlying Figs. 3c, d, 5b, and Supplementary Fig. 7 are provided as a Source Data file.

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

## Acknowledgements

The authors thank Anton Paar for the use of the Anton Paar 702 rheometer through their VIP academic research program. We also thank Dr. Ben Fabry, Dr. Paul A. Jammey, and Dr. Stefan Munster for their critical discussions of collagen network rheology. This work was supported by the National Science Foundation under Grant nos. DMR-1352043 and DMR-1606410. Research reported in this publication was supported by the National Center for Advancing Translational Sciences of the National Institutes of Health under University of Florida Clinical and Translational Science Awards TL1TR001428 and UL1TR001427. The content is solely the responsibility of the authors and does not necessarily represent the official views of the National Institutes of Health.

## Author contributions

C.D.M. and S.T.E. designed and executed the experiment, analyzed the data, created the figures, and wrote the paper. C.S.O., Y.Z., S.N., K.F.S., T.B., and K.D.S. helped perform the vital experiments. C.S.O., C.P.K., and B.S.S. developed protocol and synthesized microgels. M.S., G.L.M., and C.T.F. provided and transfected the cells, and contributed important knowledge of biology. W.G.S., D.D.T., and D.A.M. assisted with writing and editing the paper. T.E.A. conceived the experiment, analyzed the data, and wrote the paper.

## Additional information

**Competing interests:** The authors declare no competing interests

