## [Peer Review File · Nature Communications]

Editorial Note: This manuscript has been previously reviewed at another journal that is not operating a transparent peer review scheme. This document only contains reviewer comments and rebuttal letters for versions considered at Nature Communications .

REVIEWERS' COMMENTS:

Reviewer #1 (Remarks to the Author):

The authors have thoughtfully addressed all my questions and I believe this manuscript is ready for publication.

Reviewer #2 (Remarks to the Author):

The authors have greatly improved the study, particularly with a more clear focus on the advances that have been made (less developmental, more fabrication), the inclusion of new experiments to characterize the materials and interface, and a better discussion of the limitations of the work. Many of these limitations still exist, but they are now discussed clearly for the reader. The addition of printed sheets, although not the complexity that is perhaps desired, is appropriate as well.

Some minor considerations that should be addressed:

Figure 1 could be expanded with further information on material properties of both the collagen and microgel media, since these are the main focus for perturbations in the system throughout the rest of the paper. Of importance for this paper would be the influence of collagen concentration on E and the properties of the microgel media (such as unexpected G' versus yield stress linear relationship). This would set the stage for the terminology used throughout. Although may feel standard, it would tie in relationships that are mixed throughout (collagen concentrations and E ; G' and yield stress). I suggest trying to make the terminology consistent.

The data shown in Figure 4A is difficult to interpret with an arrow at the top stating variations, since although G' increases left to right, E does not. With two variables, this is not an appropriate presentation. I suggest to either remove the arrow and just report the conditions clearly written above each image, or only change one variable going from left to right (e.g., maintain collagen concentration the same, vary G').

In Figure 4B, collagen concentration is now used, rather than E . Consistency in terminology and comparisons would be helpful. My comment on Figure 1 may be helpful, I think if these simple relationships are introduced early, can stick with G' and E in rest of study.

In 4B, y-axis is "elastic modulus", but should be shear modulus or just state G' . Should check that terms are consistent throughout.

In Figure 7, the printed "sheet" gets lost in the paper. I suggest this be set as own figure with more details and discussion. What is the formulation for the collagen and printing media? How do the results for the sheet compare to the beam? Do all sheets buckle? Although the theory would be different from beams, linking to the rest of the paper is needed.

Response to Reviewers' Comments:

Reviewer #1 (Remarks to the Author):

The authors have thoughtfully addressed all my questions and I believe this manuscript is ready for publication.

We thank the reviewer for their continued support of this work.

Reviewer #2 (Remarks to the Author):

R2: *The authors have greatly improved the study, particularly with a more clear focus on the advances that have been made (less developmental, more fabrication), the inclusion of new experiments to characterize the materials and interface, and a better discussion of the limitations of the work. Many of these limitations still exist, but they are now discussed clearly for the reader. The addition of printed sheets, although not the complexity that is perhaps desired, is appropriate as well.*

Morley, et al: We are grateful to the reviewer for the helpful feedback and the support of our work. We truly appreciate the referee spending valuable time helping us to publish our manuscript. We address all the reviewer's remaining minor concerns below.

R2: *Some minor considerations that should be addressed:*

R2: *Figure 1 could be expanded with further information on material properties of both the collagen and microgel media, since these are the main focus for perturbations in the system throughout the rest of the paper. Of importance for this paper would be the influence of collagen concentration on E and the properties of the microgel media (such as unexpected G' versus yield stress linear relationship). This would set the stage for the terminology used throughout. Although may feel standard, it would tie in relationships that are mixed throughout (collagen concentrations and E; G' and yield stress). I suggest trying to make the terminology consistent.*

Morley, et al: We understand the reviewer's recommendation, and we agree that by adding the range of explored parameter space to Figure 1, we better prepare the reader to take in the rest of the manuscript. We have therefore added an additional panel to Figure 1 that specifies the range of parameter space we explore throughout the manuscript. We agree that it would help tie together the "mixed" parameters described later in the manuscript, like collagen concentration and elastic modulus.

R2: *The data shown in Figure 4A is difficult to interpret with an arrow at the top stating variations, since although G' increases left to right, E does not. With two variables, this is not an appropriate presentation. I suggest to either remove the arrow and just report the conditions clearly written above each image, or only change one variable going from left to right (e.g., maintain collagen concentration the same, vary G').*

In Figure 4B, collagen concentration is now used, rather than E. Consistency in terminology and comparisons would be helpful. My comment on Figure 1 may be helpful, I think if these simple relationships are introduced early, can stick with G' and E in rest of study.

Morley, et al: We grappled with this very issue when constructing Figure 4A, so we sincerely appreciate the reviewer's advice on this matter. Following the reviewers' recommendation, we have taken away the arrow to avoid suggesting an increase in multiple parameters across the

figure. To avoid clutter in the image and maximize space for clear images, we chose to specify the parameter details in the figure caption. We hope that the reader can use the new panel in Figure 1 (panel c) to place each set of images within the overall parameter space. We hope the reviewer agrees that this change improves the clarity of this figure in a manner consistent with the spirit of the recommendation.

R2: In 4B, y-axis is "elastic modulus", but should be shear modulus or just state G'. Should check that terms are consistent throughout.

Morley, et al: We thank the reviewer for bringing this to our attention and we have changed the y-axis to shear modulus instead of elastic modulus.

R2: In Figure 7, the printed "sheet" gets lost in the paper. I suggest this be set as own figure with more details and discussion. What is the formulation for the collagen and printing media? How do the results for the sheet compare to the beam? Do all sheets buckle? Although the theory would be different from beams, linking to the rest of the paper is needed.

Morley, et al: We understand the reviewer's observation that the "sheet" experiment is potentially understated in the manuscript. Our intention is to show it as a final demonstration, and not as another example of thoroughly explored discovery. For this reason, we couple it to the 3D printed neural crest model as a set of aspirational exhibitions of what is possible. We therefore prefer not to separate Figure 7 into two figures; we believe they are more powerful when coupled together and presented in this forward-thinking manner. However, we recognize that details of the sheet were omitted in our previous revision, so we have now added the details the reviewer requested, specifying the collagen concentration, the corresponding elastic modulus, and the shear modulus and yield stress of the surrounding microgel medium. We hope these changes provide enough information to the readers to scrutinize what we've done and potentially move forward with hypotheses of their own for future investigations into the instabilities that may arise in a diversity of different biofabricated shapes.